# CTIKG: LLM-Powered Knowledge Graph Construction from Cyber Threat Intelligence

**Liangyi Huang, Xusheng Xiao**
School of Computing and Augmented Intelligence
Arizona State University, Tempe, AZ, USA
{lhuan139, xusheng.xiao}@asu.edu

## Abstract

To gain visibility into evolving threat landscape, knowledge of cyber threats has been aggressively collected across organizations and is often shared through Cyber Threat Intelligence (CTI). While knowledge of CTI can be shared via structured format such as Indicators of Compromise (IOC), articles in technical blogs and posts in forums (referred to as CTI articles) provide more comprehensive descriptions of the observed real-world attacks. However, existing works can only analyze standard texts from mainstream cyber threat knowledge bases such as CVE and NVD, and lack of the capability to link multiple CTI articles to uncover the relationships among security-related entities such as vulnerabilities. In this paper, we propose a novel approach, CTIKG, that utilizes prompt engineering to efficiently build a security-oriented knowledge graph from CTI articles based on LLMs. To mitigate the challenges of LLMs in randomness, hallucinations and tokens limitation, CTIKG divides an article into segments and employs multiple LLM agents with dual memory design to (1) process each text segment separately and (2) summarize the results of the text segments to generate more accurate results. We evaluate CTIKG on three representative benchmarks built from 500+ real world CTI articles, and the results show that CTIKG achieves 86.88% precision in building security-oriented knowledge graphs, achieving at least 30% improvements over the state-of-the-art techniques. We also demonstrate that the retry mechanism makes open source language models outperform GPT4 for building knowledge graphs.

## 1 Introduction

In recent years, advanced cyber attacks often exploits multiple vulnerabilities and progressively penetrates the enterprise network to compromise security and confidentiality, such as the I2CE3 ransomware attack chain (Keshavarzi & Ghaffary, 2020). To effectively counter these attacks, knowledge of cyber threats has been aggressively collected across organizations and is often shared through Cyber Threat Intelligence (CTI), which models cyber threats including their actors, tactics and techniques McMillan (2013); Wagner et al. (2019). CTI has been primarily collected in the form of Indicators of Compromise (IOCs) Obrst et al. (2012); Liao et al. (2016), which are forensic artifacts of an intrusion such as malicious file/process names and IPs/domains of botnets. For example, a variety of security websites have published their own IOC reports of the Log4J vulnerability (Kim, 2022; ElastZris, 2022; Azure, 2023). Automated security detection solutions can also benefit from IOCs. For instance, the Microsoft Malicious Software Removal Tool (MSERT) employed relevant IOCs following the cyber attack on the Chilean bank regulator (Onyegbula, 2023). Additionally, CrowdStrike Falcon, Microsoft Sentinel and Microsoft Defender XDR integrate IOC as a component of their detection systems (CrowdStrike, 2023; Microsoft, 2023).

Besides structured formats like IOCs, recent studies Liao et al. (2016); Dong et al. (2019) show that the information of cyber threat delivered by traditional blacklists Group; MX is rather thin, which covers only a limited set of IOC classes (URL, domain, IP and MD5) and cannot reveal the relations between IOCs. Instead, descriptions of cyber threats, such as articles

in technical blogs and posts in forums, referred to as *CTI articles*, are more favorable for security practitioners since they provide more comprehensive descriptions of the observed real-world attacks. For example, Figure 1 shows three CTI articles that describe the malware Bedep. Based on these articles, Bedep has relationship with Angler and Magnitude, and performs click-fraud, which can not be found in professional databases. Thus, existing techniques Obrst et al. (2012); Liao et al. (2016); Catakoglu et al. (2016); Dong et al. (2019) that focus on extracting IoCs is insufficient to represent the cyber threat knowledge in CTI since they overlook the IOCs' behaviors and their relationships. To enhance the representation of the knowledge accumulated in CTI, a specialized knowledge graph that depicts domain-specific knowledge through entity-entity relationships, which has been widely used among large enterprises for data management (Yan et al., 2018) and Natural Language Processing (NLP) tasks (Liu et al., 2021), can be employed.

Constructing a knowledge graph out of CTI articles entails automatic techniques for extracting triples from the sentences within these articles, and linking these triples to form a graph. Although existing works based on pre-trained language models show promising results in extracting triples from general purpose texts, they show limited capabilities in dealing

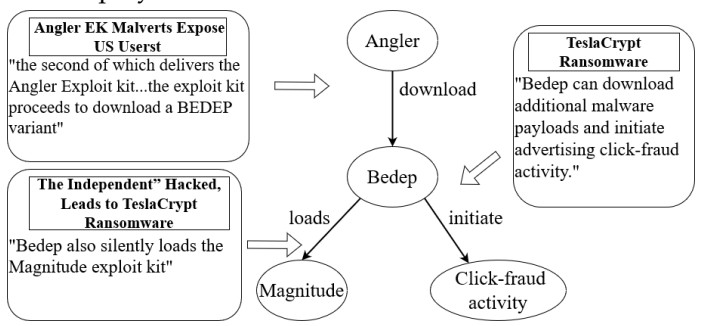

Figure 1: Correlated CTI articles for Bedep

with the complex domain-specific sentence in these CTI articles (Satvat et al., 2021; Rossiello et al., 2023). Fortunately, large language models (LLM) have recently been shown to perform much better in NLP tasks such as text comprehension, triple extraction, coreference resolution, and context reading (OpenAI, 2023a; Brown et al., 2020; 01.AI, 2023; 01-ai, 2023; Achiam et al., 2023), and thus becomes a potential solution to analyze the complex entities and relationships from CTI articles.

**Goal**. Recognizing the importance of CTI and potential of LLMs, we aim to build a novel LLM approach, CTIKG, that *effectively analyzes descriptions of security-related entities in CTI articles to build a knowledge graph* and *reveals relationships and behaviors among security-related entities across CTI articles*. In particular, the nodes in the knowledge graph represent security-related entities and the edges in the graph represent the relationships among these security-related entities across different CTI articles.

**Challenges**. We encounter several key challenges due to the limitation, hallucinations and inaccurate comprehension of LLMs:

- *Token Limitation*: Since some CTI articles have long contexts, and LLM prompts need to include chain of thoughts with examples, LLMs cannot process an entire CTI article at once due to the token limitation, and may miss important CTI information.
- *Incorrect Triple Format*: Even explicitly specified in the prompts, LLMs may not follow the required formats, and will produce extracted information as tuples containing more than two elements.
- *Incorrect Output*: LLMs may extract inaccurate information from the given text, such as the information from the given example but not the given text or wrong triple information due to misunderstanding the text.
- *Misunderstood Task*: LLMs may misunderstand the task as the task for generating subsequent text of the input CTI article and do not produce triples.

**Contributions**. To address these challenges, CTIKG performs text segmentation on CTI articles and employs multiple LLM agents with dual memory design to process text segments separately (results stored in *short-term memory*) and summarize results to produce final results (results refined in the *long-term memory*). In this way, CTIKG can effectively address the token limitation and use long chain of thoughts to guide the LLM agents to extract triples from CTI articles in every text segment. Moreover, CTIKG employs multiple LLM agents to

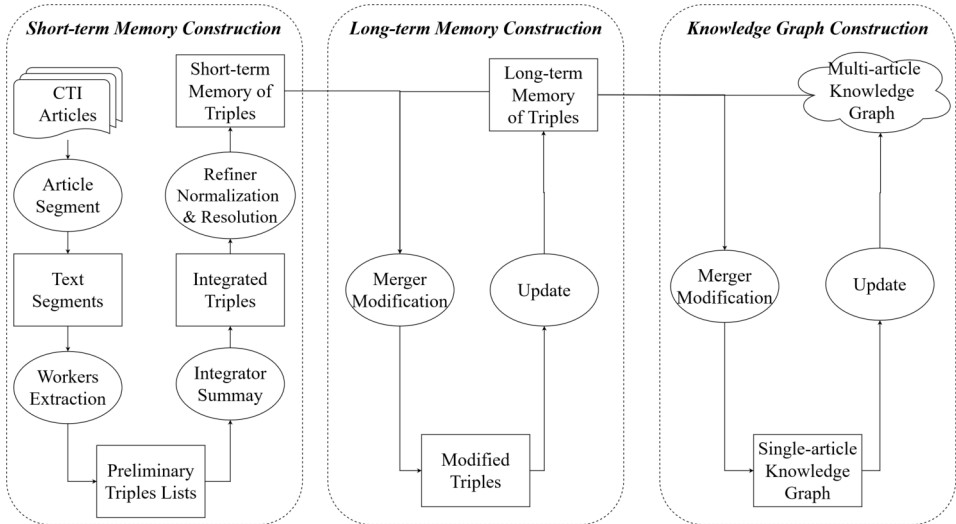

Figure 2: Architecture of CTIKG

perform triple extraction with different temperature settings and then integrates the results using another LLM agent, mitigating the hallucinations and randomness of LLMs. More importantly, CTIKG employs a checker LLM agent that instructs other LLM agents to redo their tasks if the results are incorrect, greatly improving the robustness of the constructed knowledge graph.

**Evaluations**. We evaluate CTIKG on three benchmarks constructed from 500+ real world CTI articles. For triple extraction from 255 sentence cross 13 cyber attack tactics defined by ATT&CK knowledge base (Corporation, 2022), CTIKG achieves 91.89% precision and 89.39% recall, which is better than Extractor (Satvat et al., 2021), REBEL (Cabot & Navigli, 2021) and KnowGL (Rossiello et al., 2023) by more than 10%. For knowledge graph construction from 30 CTI articles, CTIKG achieves 86.88% precision and 70.86% recall, which is at least 30% better than these three approaches in both precision and recall. We also evaluate CTIKG on 478 correlated CTI articles that describe 15 CVE entities, and show that CTIKG can averagely find 39.2 entity behaviors for each CVE entity by connecting a CTI article to 14.6 correlated CTI articles, compared to 8.6 entity behaviors that can be found by analyzing a single CTI article. We also demonstrate that local LLM models (01-ai, 2023) deployed on single GPU can outperform GPT-4 through prompt engineering, LLM agents cooperation, and chain of thoughts. Our implementation of CTIKG and the evaluation datasets are available at our project website (CTIKG, 2024). We also provide fine-tuned models at our project website (Research, 2024a). The latest model is currently based on Qwen1.5-70B (Research, 2024b).

## 2 Overview

Figure 2 shows the architecture of CTIKG, consisting of three phases:

- *Short-term Memory Construction*. CTIKG deploys multiple LLM agents to extract the knowledge within a CTI article segment in the form of triples. This phase employs four types of LLM agents: worker, integrator, refiner, and checker. The worker agents extract the knowledge from the natural language text to form the triples. The integrator agent combines the multiple results on the same text from multiple workers. The refiner agent achieves the normalization and resolution tasks on the combined triples. The checker agent inspects the triples and instructs the other agents to re-work when it detects common errors of the LLMs.
- *Long-term Memory Construction*. As CTIKG uses segment processing to overcome LLMs' token limitation, the merger agent is responsible for assembling the triples from different article segments into one long-term memory.

- *Knowledge Graph Construction*. CTIKG builds a knowledge graph using multiple articles. To ensure consistency across different articles for the same entity (e.g., a specific malware), CTIKG employs another merger agent to achieve a unified entity description.

## 3 Approach

### 3.1 Design of Triple Extraction

CTIKG extracts knowledge from article as triples in the form of *(subject, relation, object)*, where *subject* and *object* are entities and *relation* is their relation (e.g., *(Linux, is, Operating System)*).

**Short-Term Memory and Long-Term Memory**. Inspired by existing LLM studies (Park et al., 2023), CTIKG divides the article into segments and extracts the triples from each segment as the short-term memory. Meanwhile, CTIKG maintains a list of all extracted triples from different segments as the long-term memory. Figure 3 shows the algorithm for building a knowledge graph with these two memories.

**Text Segmentation**. To segment a CTI article, CTIKG first divides it into paragraphs at line breaks and merges these paragraphs into segments. For paragraphs exceeding 600 characters, CTIKG uses NLTK (Loper & Bird, 2002) to further split the paragraphs into sentences and reassemble the sentences to form segments. The rationale behind such text segmentation is shown in the Appendix A.

**Entity Types and Relationships**. Unlike the ontology-based knowledge graph, which can only extract predefined relationships and entity types, CTIKG is designed to dynamically capture all types of relationships related to any cyber threat in CTI articles. As shown in the Appendix G.3, the similarity between CVE-2012-0158 and CVE-2017-1188 was discovered based on the entity relationship "for" and "memory corruption vulnerability". Our further investigation revealed that Magniber and Cerber have a "mutual ransomware payload" relationship, and Emotet and Trickbot have a "switch to" relationship. These entity relationships are determined by the author's writing style and cannot be anticipated in advance and defined in the ontology.

### 3.2 Construction of Short-Term Memory

To construct the short-term memory of a text segment, CTIKG leverages four types of LLM agents: worker, integrator, refiner, and checker. The prompts for these agents can be found at Appendix B.

**Worker Agent**. Worker agents, told as experts in information extraction and computing, are tasked with extracting security-related data from texts (Lines 4-7). In particular, to instruct LLMs to focus on security-related data, we specify "Only extract triples that are related to cyber attacks. If a sentence does not contain any triple about cyber attacks, skip the sentence and do not include it in your output" and "Focus on malware, Trojan horses, CVEs, or hacking organizations as the subjects of the triples" in the prompts of worker agents, and provide multiple security-related examples. CTIKG deploys three worker agents with the same prompt and different temperature settings (1, 0.5, and 0.2). If at least two worker agents identify no cyber security-related triples in a segment, CTIKG skips the other agents and returns an empty result for that segment directly (Line 11). Specifically, the prompt for the worker agents contain the following parts:

- *Background*: An introduction that defines the task of triple extraction with examples.
- *Rules*: An instruction that asks the worker agent to extract only entity relations relevant to cyber attacks, and provides output format requirements.
- *Few-shots*: Three complete chat conversations, showing example inputs and responses.
- *Input*: An instruction that asks the worker agent to perform triple extraction on a given text.

**Integrator Agent**. The integrator agent extracts a final triple from the worker agents' results (Line 8). Due to the randomness of LLM (Lee et al., 2022), multiple workers may use

---

**Algorithm 1:** Single-article Knowledge Graph Construction

**Input:** Single Article $A$
**Output:** Knowledge Graph $L$

1 **Function** KnowledgeGraphConstruction($A$):
2     Split $A$ into segments $S$
3     **foreach** *segment $s$ in $S$* **do**
4         $T\_initial \leftarrow$ empty list
5         **foreach** *worker agent $w$* **do**
6             $T\_i \leftarrow WorkerAgent(s)$
7             Append $T\_i$ to $T\_initial$
8         $T\_integrate \leftarrow IntegratorAgent(T\_initial)$
9         $T\_refine \leftarrow RefinerAgent(T\_integrate)$
10         **if** *NonSecurityFeedback(T\_initial) <2* **then**
11             Skip to next segment
12         **if** *CheckerAgent(T\_refine) is True* **then**
13             **if** *long-term memory $L$ does not exist* **then**
14                 $L \leftarrow T\_initial$
15             **else**
16                 $T_{modified} \leftarrow MergerAgent(T\_refine, L)$
17                 **if** *CheckerAgent(T\_modified) is True* **then**
18                     $L \leftarrow L + T_{modified}$
19                 **else**
20                     Recall MergerAgent($T\_refine, L$) up to three times
21         **else**
22             Recall WorkerAgent(s) up to three times
23     **return** $L$

---

Figure 3: Algorithm for single-article knowledge graph construction

different triples to express the same entity relation in the text. The integrator is informed to summarize the valid triples with identical meanings and output a representative triple. Specifically, the prompt for the integrator agent contains the following parts:

- *Background*: An introduction that defines the task and provides examples to illustrate which triples are assumed to have the identical meanings.
- *Rules*: An instruction that asks the integrator agent to integrate the triples with identical meanings, and provides expected input and output format.
- *Few-shots*: A description that consists of three complete chat conversations, showing how an integrator successfully does its job with the right outputs.
- *Input*: An instruction that provides the outputs from the three worker agents as the input for the integrator agent to process.

**Refiner Agent**. The refiner performs the normalization and resolution tasks (Line 9) (Satvat et al., 2021), and leverages chain of thoughts (Wei et al., 2022) for complex task. The refiner agent takes the triples produced by the integrator agent, and performs the following tasks:

- *Simplification*: Some triple's subject and object parts have modifiers or unnecessary prefixes.
- *Splitting Complex Triples*: Some parts of the triples may contain conjunctions.
- *IOC Conversion*: Some phases may contain IOCs and should be highlighted in the triple.
- *Coreference Resolution*: The refiner agent will replace the implicit references in a triple with the actual name according to the context.
- *Normalization*: The refiner agent will standardize the relation in triples.

**Checker Agent**. Although prompts emphasize requirement several times, LLMs may still output the incorrect results. Through empirical observations of the incorrect outputs, we summarize some common patterns. The checker agent of CTIKG checks whether the outputs from other agents based on the given rules and determines whether the output is correct based on the summarized patterns of incorrect outputs (Line 12). If the output of an LLM agent is incorrect, CTIKG then instructs the LLM agent to re-perform the task. Due to resource constraints, we limit such retry to be up to three times.

## 3.3 Construction of Long-Term Memory

CTIKG uses the merger agent to further refine short-term memory triples to fit the long-term memory (Line 16). After refinement and check, CTIKG directly adds the triples into the long-term memory (Line 18). The merger agent determines whether the triples' subjects or objects in short-term and long-term memory refer to the same entities. It then modifies the corresponding subject or object in the short-term memory triple to unify representations of identical entities. Since these tasks require complex reasoning, the merger also leverages chain of thoughts to improve effectiveness (Wei et al., 2022). Specifically, the merger agent performs the following tasks:

---

**Algorithm 2:** Multi-article Knowledge Graph Construction

**Input:** Long-term memory of each CTI article $L_1, \ldots, L_n$
**Output:** Knowledge graph $G$

1 **Function** KnowledgeGraphConstruction($L$):
2     Initialize $G$ as an empty graph
3     **foreach** *CTI article $L_i$ in $L_1, \ldots, L_n$* **do**
4         **foreach** *triple $T$ in $L_i$* **do**
5             $V = \text{VectorEmbedding}(T)$
6             $N = \text{FindSimilarNodes}(V)$
7             $T_{modified} = \text{MergerAgent}(T, N)$
8             Add $T_{modified}$ as a new edge of $G$
9     **return** $G$

---

Figure 4: Algorithm for multiple CTI articles

- *Unifying Terms*: Subjects and objects that contain partially identical terms and refer to the same entities are unified by removing prefixes, suffixes, or modifiers.
- *Modifier Removal*: Specific names, such as those of malware or hacker organizations, should have all unnecessary modifiers removed.
- *Hallucination Restriction*: The creation of triples through LLM's imagination is strictly prohibited.

## 3.4 Building Knowledge Graph

Based on the long-term memory of each CTI article, CTIKG builds a knowledge graph using the subjects and the objects as nodes and the relations as edges. Figure 4 shows the algorithm for knowledge graph construction. Specifically, CTIKG use triples from first article to build the a basic graph, then gradually incorporates triples from other CTI articles. To maintain consistency of entity names across articles, CTIKG leverages another merger agent. CTIKG first uses the RoBERTa model function (Liu et al., 2019) to achieving the text embedding of subject and object (Line 5). For each triple to be added, CTIKG searches five most similar nodes in the knowledge graph based on the embedding vector similarity (Line 6). These five node names are used by the merger agent to refine the triple (Line 7). Finally, CTIKG connects the refined triple to the knowledge graph (Line 8).

## 4 Evaluation

In the evaluations, we aim to answer the following research questions:

- RQ1: How effectively can CTIKG extract triples from CTI articles?
- RQ2: How effectively can CTIKG generate knowledge graphs for CTI articles?
- RQ3: How effectively can CTIKG reveal relationships of security-related entities from correlated CTI articles?

### 4.1 Evaluation Setup

**Implementation**. We implement CTIKG ( 20K characters prompts) on a Ubuntu 20.04.6 server with AMD 5955WX CPU, RTX 6000 GPU and 128GB memory upon VLLM, OpenAI, Networkx, Stanford NLP tool suite, and NLTK (Kwon et al., 2023; OpenAI, 2023b; developers, 2023; Lab, 2003; Loper & Bird, 2002). We choose Yi-34B-4bits model as the core LLM model. We provide more details on LLM selection in Appendix D

**Evaluation Subject**. We curate a large dataset with 72,538 CTI articles from security-related websites and cyber threat knowledge base, and construct three benchmarks:

- *RQ1 Sentence Benchmark*: For each enterprise tactic defined by the Mitre ATT&CK (Corporation, 2022), we select over 20 representative sentences from the CTI articles, resulting in 255 CTI sentences with 699 manually extracted triples.
- *RQ2 CTI Article Benchmark*: We category CTI articles into three types: standalone (highlighting a single cyber threat), chain (introducing multiple interlinked cyber threats), and overview (introducing multiple cyber threats that are not directly related) categories. Ten articles were selected from each category for manual analysis, resulting in 30 knowledge graphs with a total of 315 edges.
- *RQ3 Correlated CTI Article Benchmark*: Using the Stanford NLP tool (Lab, 2003), we perform entity name extraction on the all CTI articles, identify 15 most common CVE entities, and construct a knowledge graph based on the 478 articles containing these entities. The average length of an article is 5,200 characters. The graph has 27,070 unique nodes, 26,475 edges and 7,623 unique edges. In total, 39.45% of the nodes are related to computer science topics, and 51.94% of the edges are related to computer science topics.

We provide more details of the dataset in Appendix E.

## 4.2 RQ1: Effectiveness of Triple Extraction

To evaluate CTIKG's triple extraction, we compare CTIKG with three baseline approaches and two variants of CTIKG powered by GPT-4 (Achiam et al., 2023) and GPT-3.5 (OpenAI, 2023a). The evaluation is conducted on the triple extraction benchmark constructed from the CTI article dataset.

**Baseline Approaches**. We compare CTIKG with two state-of-the-art entity relation extraction approaches: REBEL (Cabot & Navigli, 2021) and KnowGL (Rossiello et al., 2023), both of which are transformer-based seq2seq models for generic text. We also compare CTIKG with a secure text-oriented approach, Extractor (Satvat et al., 2021), which are based on BERT (Devlin et al., 2018) and BiLSTM (Hameed & Garcia-Zapirain, 2020). We also use two variants of CTIKG (CTIKG based on GPT-4 and GPT-3.5) to evaluate the impact of the underlying LLM models. These two variants of CTIKG use the same prompts, except for the checker agent and the corresponding retry mechanism. This is due to the fact that OpenAI only provides the paid API, and applying the retry mechanism will result in significant expenses.

**Effectiveness Comparison**. Table 1 shows the precision and the recall of CTIKG and the baseline approaches. Overall, CTIKG with the YI model achieves the best performance with 91.89% precision and 89.39% recall, and CTIKG with GPT-4 achieves the second best performance: 84.26% precision and 83.66% recall. The precision and the recall of CTIKG with GPT-3.5 are only about 77.88%. Meanwhile, two approaches for general texts, REBEL and KnowGL, completely fail the task of triple extraction on the security-related sentences, achieving both precision and recall below 21%. Extractor achieves a slightly worse performance with 82.56% precision and 79.35% recall. For specific cyber attack tactics, CTIKG achieves the best performance ($> 94\%$) for the tactics of Persistence, Discovery, and Lateral Movement, and worst for the tactics of Impact (about 84%). The second best approach, CTIKG with GPT-4, has a larger variant for different tactics, achieving only 71.59% for the tactics of Initial Access.

While GPT-4 is a better-trained model with top MMLU scores, thus CTIKG with GPT-4 suffers from the randomness and hallucination due to lacking of checker agent and retry mechanism. CTIKG with GPT-3.5 has the lowest MMLU score model, and lacking the checker agent and retry mechanism, achieves the worst results among these three versions of CTIKG. While Extractor is competitive with CTIKG with GPT-4 in terms of accuracy due to its well-designed internal components and multi-step pipeline, its text comprehension capability is less powerful than that of LLMs. When processing sentences like *"The first thread is responsible for finding the CHD in the process, writing the results to a file, and preparing the files for exfiltration"*, Extractor cannot find the relationship between *the first thread* and *the file*. This makes its recall lower than CTIKG with YI and GPT-4.

Table 1: Comparison of CTIKG and baseline approaches in triple extraction

| Tactics | CTIKG | | CTIKG with GPT-4 | | CTIKG with GPT-3.5 | | Extractor | | REBEL | | KnowGL | | Triple Source |
|---|---|---|---|---|---|---|---|---|---|---|---|---|---|
| | Precision | Recall | Precision | Recall | Precision | Recall | Precision | Recall | Precision | Recall | Precision | Recall | # of Sentence |
| Initial Access | **91.89%** | 85.61% | 71.59% | 69.84% | 81.77% | 82.20% | 73.65% | 75.00% | 17.86% | 17.46% | 4.55% | 1.52% | 22 |
| Execution | **88.27%** | 82.16% | 82.23% | 82.01% | 65.35% | 67.61% | 84.98% | **83.75%** | 17.44% | 17.25% | 13.26% | 5.87% | 47 |
| Persistence | **95.37%** | **98.15%** | 74.25% | 70.99% | 77.47% | 76.54% | 83.68% | 80.67% | 33.33% | 30.86% | 24.07% | 12.35% | 29 |
| Privilege Escalation | **95.03%** | 85.83% | 72.51% | 71.18% | 71.02% | 67.36% | 91.52% | **87.39%** | 21.53% | 10.56% | 18.06% | 5.56% | 26 |
| Defense Evasion | **87.39%** | 85.61% | 93.56% | **88.06%** | 78.17% | 80.00% | 79.85% | 82.64% | 16.67% | 9.72% | 16.11% | 6.90% | 32 |
| Credential Access | **92.01%** | **88.10%** | 77.99% | 77.62% | 74.60% | 76.90% | 82.99% | 76.96% | 22.98% | 16.95% | 15.69% | 10.19% | 37 |
| Discovery | **95.91%** | **94.22%** | 88.33% | 87.50% | 76.89% | 83.75% | 79.26% | 75.97% | 10.00% | 9.95% | 8.33% | 10.22% | 34 |
| Lateral Movement | **94.62%** | **95.26%** | 89.81% | 88.89% | 79.80% | 80.21% | 88.04% | 80.07% | 31.25% | 17.36% | 26.39% | 14.93% | 29 |
| Collection | **94.76%** | **92.94%** | 85.50% | 87.90% | 82.83% | 84.92% | 85.30% | 76.22% | 28.33% | 17.40% | 17.89% | 11.11% | 43 |
| Command and Control | 89.08% | 90.04% | **95.05%** | **96.17%** | 70.23% | 76.35% | 79.00% | 76.71% | 23.10% | 14.31% | 15.74% | 6.20% | 39 |
| Exfiltration | **93.84%** | **90.62%** | 88.01% | 86.63% | 82.37% | 87.71% | 89.12% | 84.57% | 16.07% | 10.20% | 6.59% | 6.82% | 47 |
| Impact | 84.49% | 84.10% | **92.27%** | **97.10%** | 71.92% | 71.01% | 73.36% | 72.27% | 23.41% | 18.62% | 17.39% | 10.94% | 27 |
| Average | **91.89%** | **89.39%** | 84.26% | 83.66% | 76.04% | 77.88% | 82.56% | 79.35% | 21.83% | 15.89% | 15.34% | 8.55% | 34.44 |

**Error Analysis**. Since LLMs are essentially text generation models, the next word generated is dominated by probability (OpenAI, 2023a). We summarize some LLM-specific errors in triple extraction as follows:

- *Hallucination*: Hallucination is the most common problem. For example, for sentence *"Both apps upload users phone book to remote server and use it for SMS spam"*, GPT-4 generates new non-existent sentence and incorrect triple ⟨ AdwareY, is linked to, GhostNet⟩.
- *Misunderstood Task*: Another common error is misunderstood task. For example, for the sentence *"This image file exists on the third page of the document, so the user would have to scroll down in the document to this third page to get the SWF file to run"*, CTIKG with GPT-4 may outputs the incorrect triple ⟨ Formbook, is, malware⟩ based on the few-shot example. 13.42% and 6.95% of the outputs of CTIKG with GPT-3.5 and GPT-4, respectively, incorrectly use few-shot sample text due to lack of checker agent.
- *Text Comprehension Ability*: The level of text comprehension ability has a great impact on the outputs. CTIKG with GPT-3.5 often misunderstands the content, while CTIKG with the YI model and GPT-4 are much less likely to do so. For the sentence *"Octopus uses wmic for local discovery information."*, only CTIKG with GPT-3.5 produced the incorrect result ⟨ Octopus, uses, information⟩.

Recent studies show that evaluations using LLMs can match or surpass the accuracy of manual inspection (McAleese et al., 2024; Zheng et al., 2024; Kenton et al., 2024). In addition to manual inspection, we use an automatic evaluator based on GPT-4, which has an accuracy rate of 93%, to evaluate the correctness of triples and categorize their error types using a few-shot learning and debating mechanism. The results demonstrate that CTIKG, when utilizing the YI model and checker agent, extracts fewer hallucinated triples compared to CTIKG based on GPT-4 and GPT-3.5. This finding demonstrates that the checker agent effectively filters out hallucinated triples. Additionally, the number of triples with incorrect relationships is lower than that produced by CTIKG using the other two LLM models. Hallucinations often cause other triples within the same sentence to be incorrect. Consequently, the checker, despite not being specifically designed to address such errors, can reduce the number of incorrect triples by filtering out hallucinated triples. The detailed figure is shown in Appendix F.2.

**Impacts of Sentence Complexity**. We measure the complexity of a sentence using the number of ground truth triples and word count. As shown in Figure 6, with the complexity increases, the precision of CTIKG with the YI model improves while its recall decreases, and thus the overall effectiveness does not change much. It is mainly due to the check agent that leverages retry to minimize errors. CTIKG with GPT-4 achieves similar performance with the complexity increases but has a larger variance due to the randomness of GPT-4. Both Extractor and CTIKG with GPT-3.5 achieve worse results when the complexity increases. Although the syntax of these sentences is relatively simple, it requires Extractor to be able to detect a relationship between two entities far apart. For example, Extractor fails to extract triples from the sentence "The overall purpose of Cannon is to use several email accounts to send system data (system information and screenshot) to the threat actors and to ultimately obtain a payload from an email from the actors". CTIKG with the YI model and CTIKG with GPT-4 have better context reading capabilities, which allows them to find these relationships correctly. Figure 7 shows a similar trend for Extractor: with the increase of the words in a sentence, Extractor performs worse. Overall, CTIKG with the YI model and CTIKG with GPT-4 achieve the most robust results for sentences of different complexity. We notice that the performance of CTIKG with GPT-4 does not degrade as the number of triples or the

Table 2: Comparison of CᴛɪKG and baseline approaches in knowledge graph construction

| Type | CᴛɪKG | | Extractor | | REBEL | | KnowGL | |
|---|---|---|---|---|---|---|---|---|
| | Precision | Recall | Precision | Recall | Precision | Recall | Precision | Recall |
| Standalone | **90.21%** | **79.63%** | 50.83% | 34.03% | 31.67% | 6.11% | 39.58% | 9.81% |
| Chain | 82.38% | **62.89%** | **66.00%** | 38.67% | 46.67% | 16.67% | 0.00% | 0.00% |
| Overview | **88.04%** | **70.08%** | 50.97% | 50.13% | 15.64% | 5.05% | 5.28% | 2.27% |
| Average | **86.88%** | **70.86%** | 55.93% | 40.94% | 31.32% | 9.28% | 14.95% | 4.03% |

number of words within a sentence increases. This is mainly due to GPT-4's larger token limit, which is $131,072$ tokens, larger than GPT-3.5 and the YI model's default $4,096$ tokens.

**Entity Types and Relationships**. We have manually 699 cyber security-related triples of the RQ1 sentence benchmark, and fount out that 51.96% of nodes cannot be categorized into common entity types such as malware names, and 54.93% of edges cannot be categorized into computer science-related relationship types. These results show that most of the entities and their relationships cannot be easily anticipated in advance and defined in the ontology.

### 4.3 RQ2: Knowledge Graph Construction

To evaluate CᴛɪKG's knowledge graph construction, we compare CᴛɪKG with the three baselines: Extractor (Satvat et al., 2021), REBEL (Cabot & Navigli, 2021), and KnowGL (Rossiello et al., 2023) on the knowledge graph benchmark. Since we demonstrated that CᴛɪKG performs better than CᴛɪKG with GPT-4 and GPT-3, we no longer include them in RQ2.

**Effectiveness Comparison**. As shown in Table 2, CᴛɪKG achieves the best precision (86.88%) and recall (70.86%), which is at least 73% better than the second best approach, Extractor, in both precision and recall. Extractor achieves a precision of 55.93% and a worse recall of 31.32%. Both REBEL and KnowGL perform poorly on these security-related CTI articles, with precision and recall below 35%. This is consistent with their triple extraction effectiveness since inaccurate triple extraction will cause the built knowledge graph to be even more inaccurate.

Across article types, CᴛɪKG performs best with standalone articles and least effectively with chain articles. This variance is due to the complexity of chain articles, which detail multiple interrelated threat entities, making them more difficult to analyze than standalone articles, which focus on a single cyber threat and maintain cohesion on a topic. Despite these challenges, CᴛɪKG's precision and recall significantly outperform baseline methods, with improvements of at least 24.8%. We provide more details on the impacts of article type in Appendix C.

**Error Analysis**. By inspecting the final results of CᴛɪKG, and the intermediate results of the short-term memories and the long-term memories, we found that besides those common errors observed in triple extraction, CᴛɪKG also suffered from the following problems:

- *Length limit*: Although the LLM context length limit is extended to $16,384$ tokens, CᴛɪKG may still need more space during the chain of thoughts process (Wei et al., 2022). If there are too many triples in the short-term memory, the merger agent will reach the length limit when performing the chain of thoughts and output unfinished results. This limitation cannot be easily addressed by performing retry.
- *Incorrect Coreference Resolution*: If an article segment uses a demonstrative pronoun instead of a specific name throughout the text, the merge agent may incorrectly replace those pronouns with another specific name from the long-term memory. This error can be mitigated by recording the original sentence for each triple, but this method will make the problem of length limit worse.

### 4.4 RQ3: Revealing Entity Relationships Across Correlated CTI Articles

In this RQ, we compare the knowledge graph constructed from correlated CTI articles with the knowledge graph constructed from a single CTI article, and use the number of edges connected to a specific node in the knowledge graph as a metric to represent the number of that entity's behaviors described within the graph.

**Effectiveness Comparison**. For the 15 CVE entities in our RQ3 benchmark, the knowledge graph based on a single article has an average of only 8.6 edges per CVE entity, representing 8.6 entity behaviors extracted from the article. The multi-article knowledge graph has averagely 39.2 entity behaviors per CVE entity. It indicates that by integrating the knowledge from correlated CTI articles, each CVE entity gains an additional 30.6 entity behavior descriptions, which is a 428.76% improvement. On average, each CVE is connected to 14.6 other articles, with 2.1 additional entity behaviors found by connecting to a correlated article. Appendix G shows the detailed effectiveness data for each CVE, and a corresponding case study.

## 5 Discussion

**Retrieval Augmented Generation**. We used Retrieval Augmented Generation (RAG) (Jeong, 2023) to dynamically generate examples for each prompt. However, since LLM sometimes uses these examples as part of the input, we cannot predict the wrong output that may occur when hallucinations occur when using RAG. Currently, we use fixed examples to eliminate the hallucination phenomenon.

**Performance and Transferability** The current CTIKG based on YI-32B processes an article in 5 minutes on dual RTX A6000. The newly released model, LLaMA3-8B, has shown better performance with a smaller model size according to LMSYS Chatbot Arena Leaderboard (Chiang et al., 2024). Since CTIKG can seamlessly switch to any LLM model as the backend, tool with LLaMA3-8B takes only 50 seconds to process an article. In the long run, the CTIKG with improved LLMs will achieve better performance with more advanced GPUs.

**Other Domains** While it is not within our current focused scope, in future work, we plan to verify the effectiveness of CTIKG with adaptation to other domain knowledge in other domains like SciERC (Luan et al., 2018).

## 6 Related Work

**Knowledge Graph Construction.** Nidhi et al. Rastogi et al. (2020) propose an ontology to generate knowledge graphs for malwares. Aritran et al. Piplai et al. (2020) executes malware and record the behavior in a knowledge graph and merge with knowledge extracted from CTI blogs. Shenqi et al. Qin & Chow (2019) and Anders et al. Høst et al. (2023) build CVE-related knowledge graphs based on the NVD database's context. Rossiell et al.Rossiello et al. (2023) introduce a End-to-End knowledge graph construction approach for general text. Some recent research Hu et al. (2024); Yao et al. (2023) also use LLM knowledge graph construction. CodeKGC Bi et al. (2024), a knowledge graph construction approach based on LLMs, can extract only pre-defined entity relationships and handle each sentence without considering the context of other sentences in the article, while CTIKG can automatically uncover different types of security-oriented entity relationships and process the whole article with the help of the dual memory design.

## 7 Conclusion

We have proposed CTIKG, which employs multiple LLM agents and dual memory design to build a knowledge graph from CTI articles with high effectiveness. CTIKG divides a long CTI article into text segments and processes each text segment separately using multiple LLM aganets with different temperature settings for mitigating the randomness of LLMs. CTIKG then summarize the results of the text segments to generate more accurate results. Our evaluations on two representative benchmarks derived from real world CTI articles demonstrate the superiority of CTIKG over the state-of-the-art approaches.

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

## A   Research on Text Segmentation Parameter

We used a data-driven method to set the text segment size, analyzing $13,385$ CTI-related sentences with an average length of 141 characters. Open source LLMs, such as the YI model (01-ai, 2023), accurately handle up to 600 characters or about 4 sentences. Hence, we limit segments to 600 characters.

## B   Prompts for LLM Agents

### B.1   Prompt for Worker Agent

.

---

**User**: "As an AI trained in entity extraction and relationship extraction. You're an advanced AI expert, so even if I give you a complex sentence, you'll still be able to perform the relationship extraction task. A triple is a basic data structure used to represent knowledge graphs, which are structured semantic knowledge bases that describe concepts and their relationships in the physical world. A triple must has three elements: *[Subject, Relation, Object]*. In entity extraction, you follow those rules:"

**Rule 1**: "Only extract triples that are related to cyber attacks. If a sentence does not have any triple about cyber attacks, skip the sentence and do not print it in your output."

**Rule 2**: "Make sure your results is a python dictionary format. One example is *[subject1, relation1, object1]*."

---

**Preset assistant responses**: "I got it."

**User**: "Here is one sentence from example article:attempts to infiltrate target networks through various means of intrusion: watering hole websites, vulnerability scans of network services on the internet, and brute-force/dictionary login attempts."

**Preset assistant responses**: "*[Leafminer, attempts to infiltrate, target networks], [Leafminer, use, watering hole websites], [Leafminer, use, vulnerability scans of network services on the internet],[Leafminer, use, brute-force], [Leafminer, use, dictionary login attempts].*"

**User**: "Here is another sentence from example article: Kismet is also a powerful tool for penetration testers that need to better understand their target and perform wireless LAN discovery."

**Preset assistant responses**: "*[Kismet,is a powerful tool for, penetration testers], [testers, understand, their target], [testers, perform, wireless LAN discovery].*"

**User**: "Here are my new sentence, extract all possible entity triples from it. Now, I start to give you sentence. *[Input Text]* .Now, my input text are over. You MUST follow the rules I told you before."

## B.2 Prompt for Integrator Agent

**User**: "You are responsible for combining the three different entity extraction results from three different assistants extracting from the same sentence into one. The final results is a python list format. When you encounter multiple triples with the same meaning but slightly different words, choose the one that expresses it most succinctly and do not add the others to your final result. Just answer me the final python dictionary with triple format without any other words."

## B.3 Prompt for Refiner Agent

**User**: "You play the role of an entity extraction expert and modify/simplify/split the text (extracted multiple triples) in the entity extraction result I gave you. according to the following rules. A triple is a basic data structure used to represent knowledge graphs, which are structured semantic knowledge bases that describe concepts and their relationships in the physical world. Here are all rules you must follow."

**Rule 1**: "If the subject or object in a triple contains pronouns such as it, they, malware, trojan, attacker, ransomware, or group, replace them with a specific name as much as possible according to the context."

**Rule 2**: "Focus on malware, Trojan, CVE, or hacking organization or as the subject of the triples, if a subject with malware or Trojan or CVE or hacking organization is found and has additional suffixes, remove the suffixes."

**Rule 3**: "Split a complex triple into multiple simpler forms. For example, *[Formbook and XLoader, are, malware] should be split into [Formbook, is, malware] and [XLoader,is, malware].*"

**Rule 4**: "If the [subject, relation] in a triple can be formed into a new [subject, relation, object] triple, create a new triple while keeping the original one."

**Rule 5**: "If the object can be simplified to a more concise, generic expression, create a new triple while keeping the original one. For example, *[Formbook, save, XLoader in desktop]* must has a new triple *[Formbook, save, XLoader]* due to the object XLoader in desktop can be simplified to XLoader."

**Rule 6**: "Simplify the subject, object, and relation into a more concise, generic expression."

**Rule 7**: "When you encounter a plural or past tense form, convert it to singular or present tense. For example, [Windows users] should be converted to [Windows user]."

**Rule 8**: "When you encounter an MD5, registry, path, or other identifier that contains prefixes, remove them. For example, *[md5 5a23c3cb225ad54175e810bc653f59dd]* should be simplified to *[5a23c3cb225ad54175e810bc653f59dd]*."

**User**: "Here is my entity extraction result: *[input text]*. Now, you apply the rules I told you before. First write down your though, think it step by step. In the end, you musttell me the final new entity extraction result."

### B.4 Prompt for Refiner Agent

**User**: "You play the role of an entity extraction expert and modify/simplify/split the text (extracted multiple triples) in the entity extraction result I gave you. according to the following rules. A triple is a basic data structure used to represent knowledge graphs, which are structured semantic knowledge bases that describe concepts and their relationships in the physical world. Here are all rules you must follow."

**Rule 1**: "If the subject or object in a triple contains pronouns such as it, they, malware, trojan, attacker, ransomware, or group, replace them with a specific name as much as possible according to the context."

**Rule 2**: "Focus on malware, Trojan, CVE, or hacking organization or as the subject of the triples, if a subject with malware or Trojan or CVE or hacking organization is found and has additional suffixes, remove the suffixes."

**Rule 3**: "Split a complex triple into multiple simpler forms. For example, *[Formbook and XLoader, are, malware] should be split into [Formbook, is, malware] and [XLoader,is, malware]."*

**Rule 4**: "If the [subject, relation] in a triple can be formed into a new [subject, relation, object] triple, create a new triple while keeping the original one."

**Rule 5**: "If the object can be simplified to a more concise, generic expression, create a new triple while keeping the original one. For example, *[Formbook, save, XLoader in desktop]* must has a new triple *[Formbook, save, XLoader]* due to the object XLoader in desktop can be simplified to XLoader."

**Rule 6**: "Simplify the subject, object, and relation into a more concise, generic expression."

**Rule 7**: "When you encounter a plural or past tense form, convert it to singular or present tense. For example, [Windows users] should be converted to [Windows user]."

**Rule 8**: "When you encounter an MD5, registry, path, or other identifier that contains prefixes, remove them. For example, *[md5 5a23c3cb225ad54175e810bc653f59dd]* should be simplified to *[5a23c3cb225ad54175e810bc653f59dd]*."

**User**: "Here is my entity extraction result: *[input text]*. Now, you apply the rules I told you before. First write down your though, think it step by step. In the end, you musttell me the final new entity extraction result."

### B.5 Prompt for Merger Agent

**User**: "You are a triples integration assistant. Triple is a basic data structure, which describes concepts and their relationships. But you can only see part of the article at a time. In order to record all the triples from a article, you have the following long-term memory area to record the triples from the entire article. long-term memory stores information on the article parts you have already read.

-The start of the long-term memory area-

-Triples will be added here-

-The end of the short-term memory area-

Second, you now see a part of this article. Based on this part, you already extract such triples and place them in your short-term memory:

-The start of the short-term memory area-

-Triples will be added here-

-The end of the short-term memory area-

Third, now review your long-term memory and short-term memory. Modify the short-term memory into a new short-term memory. You should follow following rules to modify triples in short-term memory to make them consistent with triples in long-term memory. You should write down how you use the rule to modify the triples in short-term memory. In additional, if you find any triples in long-term memory also need to modify based on the rule, you should also write down how you use the rule to modify the triple in long-term memory, and then add new modified triples in short-term memory as a new triple.

**Rule 1**: "You notice that in these triples, some triples have subjects and objects that contain partially identical terms and refer to the same specific nouns, but these specific nouns have prefixes/suffixes/modifiers that make them not identical. You should delete the prefixes/suffixes/modifiers and unify them into the same specific nouns."

Before rule: *[the Formbook, is designed to run as, a deleter] [Formbook sample, is designed to run as, one-time encryptor]*

After rule: *[Formbook, is designed to run as, a deleter] [Formbook, is designed to run as, one-time encryptor]*

Explanation: The words the Formbook and Formbook sample refer to the same entity, so they are unified to use the exact same subject Formbook for consistency.

**Rule 2**: "Be especially careful that when you meet specific names of malware,CVE, Trojans, hacker organizations, etc., always use their specific names and remove the prefixes/suffixes/modifiers."

Before rule: *[Malware Formbook, is, malware]*

After rule: *[Formbook, is, malware]*

Explanation: The word Formbook is a specific name of malware, so it should be used as the subject of the triple and the prefix Malware should be removed.

**Rule 3**: "Don't add non existing triples to your new short-term memory."

Suppose you find in long-term memory: *[the malware, download, Leafminer]* and in short-term memory: *[Formbook, is, malware]*. You cannot add a new triple in new short term memory: *[Formbook, download, Leafminer]*. Because you don't have evidence that the malware in the long-term memory specifically refers to Formbook.

**Rule 4**: "Don't add non existing triples that don't exist in long-term memory or short-term memory to your new short-term memory. You should add triples from long-term memory or short-term memory to your new short-term memory, not from your imagination and self creation"

**Rule 5**: "Don't add any example word like Formbook, XLoader, Leafminer, FinSpy, Kismet in your new short-term memory area, they are just example words not the real triples in the long term memory area or short term memory area."

**Rule 6**: "new short-term memory area must be started with The start of the new short-term memory area and ended with The end of the new short-term memory area."

**Preset assistant responses**: "Yes, I understand and totally follow the rules."

**User**: "-The start of the long-term memory area-

[HackerAAA, disrupts, network operations], [HackGroup CCC, targets, financial institutions], [Malware BBB, steals, personal data], [CyberAttack 2042, compromised, database systems], [A sample of BBB, save, a payload]

-The end of the long-term memory area-

-The start of the short-term memory area-

[The HackerAAA Trojan, created by, a sophisticated hacker group], [CCC hackers, launched, series of phishing campaigns], [BBB malware program, encrypts, files for ransom], [Attack in 2042, led to, sensitive leaks],[BBB, run, payload]

-The end of the short-term memory area-"

**Preset assistant responses**: "Modifications required based on the rules: Apply Rule 1 and Rule 2: Unify subjects and objects with partially identical terms and remove any prefixes/suffixes/modifiers, particularly for specific names like malware and hacker groups. Modified short-term memory:

*[The HackerAAA Trojan, created by, a sophisticated hacker group] → [HackerAAA, created by, sophisticated hacker group] [CCC hackers, launched, series of phishing campaigns] → [HackGroup CCC, launched, phishing campaigns] [BBB malware program, encrypts, files for ransom] → [Malware BBB, encrypts, files for ransom] [Attack in 2042, led to, sensitive leaks] → [CyberAttack 2042, led to, sensitive data leaks]*

I also find a triple in long-term memory that needs to be modified based on Rule 1 and Rule 2:

*[A sample of BBB, save, a payload] → [BBB, save, payload]*

Apply Rule 3 and 4 and 5: We must ensure we do not create non-existing triples when merging the memories. After applying the modifications, we now create a new short-term memory area:

-The start of the new short-term memory area-

*[HackerAAA, created by, sophisticated hacker group], [HackGroup CCC, launched, phishing campaigns], [BBB, encrypts, files for ransom], [CyberAttack 2042, led to, sensitive data leaks], [BBB, run, payload], [BBB, save, payload]*

-The end of the new short-term memory area-"

**User**: "Good. Now, lets switch to another article.

-The start of the long-term memory area-

*[Input Text]*

-The end of the long-term memory area-

-The start of the short-term memory area-

*[Input Text]*

-The end of the short-term memory area-

Now, follow the rules. Write down how you use the rule to modify the triples in short-term memory. Then, write down new short-term memory which must be started with -The start of the new short-term memory area- and ended with -The end of the new short-term memory area-"

### B.6 Prompt for Checker Agent

> **User**: "You are responsible for check the result from another AI agent. You job is to check whether the result contains following words.*[Input Text]*. The text may in different order or different form or obfuscated. Now,here is that result *[Input Text]*. You answer should just be 'No error' or 'I found error'"

## C Impacts of Article Type on Knowledge Graph Construction

We evaluate the performance of CTIKG on different article types. In general, the standalone type is the simplest, where the author describes only one security threat, such as a malware or a CVE, in an article. As a result CTIKG maintains a high recall. The reason for the lower recall is that some authors tend to mention the specific name of the threat only at the beginning of the article, and use the indicative pronouns to refer to it in all other parts of the article. For example, the article *"Unix-privesc-check - Unix/Linux User Privilege Escalation Scanner"* Darknet (2015) mentions the name Unix-privesc-check only twice in the entire article. For CTIKG, even if the specific name of it is not in the short-term memory, it still obtains its name and completes the knowledge graph by retrieving the name from the long-term memory. As for Extractor, even though it also performs coreference resolution, quite a number of edges in the knowledge graph still link to *"it"* node.

The overview article tends to describe different cyber threats in different parts. For example, the article *"A Brief History of Malware: Part Three (1993-1999)"* Crawley (2016) describes different malware in different paragraphs. For this type of article, CTIKG completes the construction of the whole article's knowledge graph through the direct stacking of the short-term memories from article segments.

For both CTIKG and Extractor, the chain type is definitely the most challenging to process. This type of articles often describe how a malicious actor or team utilizes multiple tools to reach an intrusion and complete the final task, often stealing data or ransomware. For example, the article *"IR Team Investigations Uncover eCrime Use of Nation-State Attack Methods"* Etheridge (2018) describes how an attacker can start an intrusion at the beginning and maintain the persistence mechanism to eventually install ransomware on the target computer by using multiple attack methods. For each attack method, the article provides some information that can be extracted as triples. When the long-term memory contains a large number of information about different cyber threats, the merger agent of CTIKG is more likely to produce inaccurate results, making the final result inferior to the result of the other two types.

## D LLM Selection

We prefer open source LLMs that can be deployed on our own server. We finally chose the YI 34B model (01-ai, 2023) that achieved top scores on the Alpaca Eval Leaderboard (Li et al.) and the MMLU benchmark (Hendrycks et al., 2020). We also attempted to use LLaMa 2 (Touvron et al., 2023). However, in early experiments, even the largest LLaMa 2 70B model cannot strictly follow CTIKG's prompt instructions, and result in an accuracy of merely 30%. This phenomenon is consistent with its lower score on the MMLU benchmark, and thus CTIKG uses only the YI 34B model for CTIKG.

## E Setup of Evaluation Dataset

We collect $88, 131$ CTI articles from 67 websites and 672 articles from the ATT&CK knowledge base (Corporation, 2022). Since some articles contain very short descriptions, such as just listing URLs or file addresses, we further apply NLP tools to filter out them and obtain $72, 538$ meaningful articles. These articles have average 52 sentences, where each sentence has an average of 15 words or 100 characters (Figure 5). With CTIKG capacity to process 4 sentences at a time, most articles require processing in 10 segments.

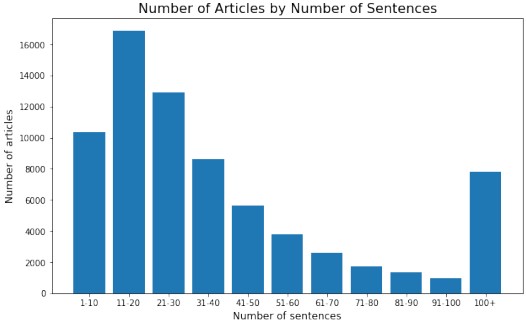

Figure 5: Distribution of articles by sentence length

# F    Statistics of RQ1 Effectiveness Comparison

## F.1    The Impact of Text Complexities

Figure 6 and Figure 7 show the accuracy of models at different text complexities.

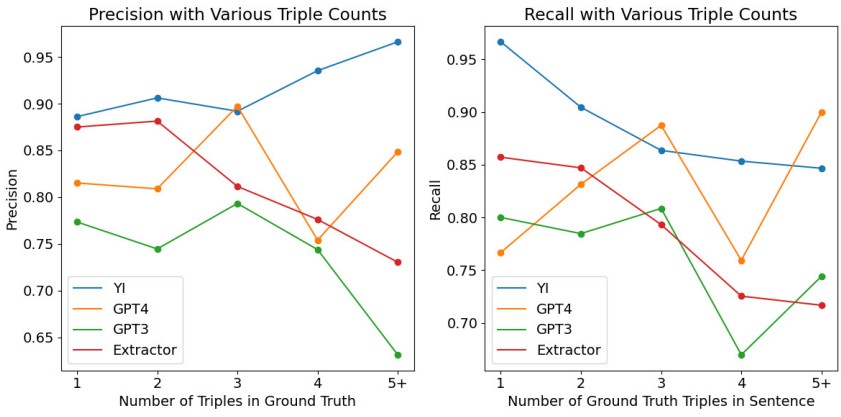

Figure 6: Evaluation based on varying sizes of triples in one sentence

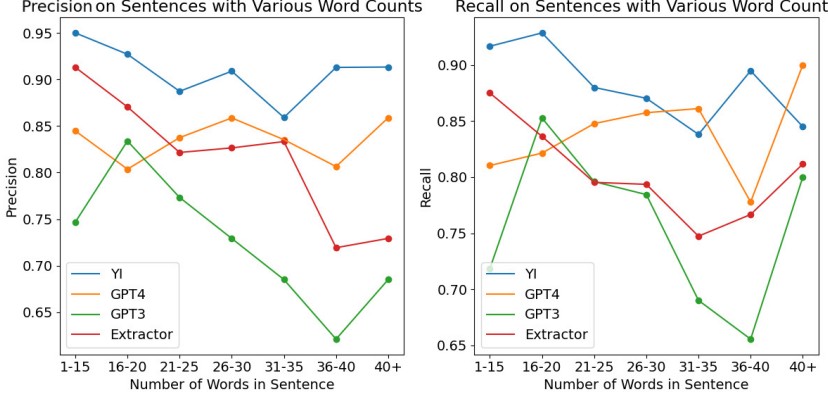

Figure 7: Evaluation based on varying number of words in one sentence

### F.2 Analysis of Error Triples

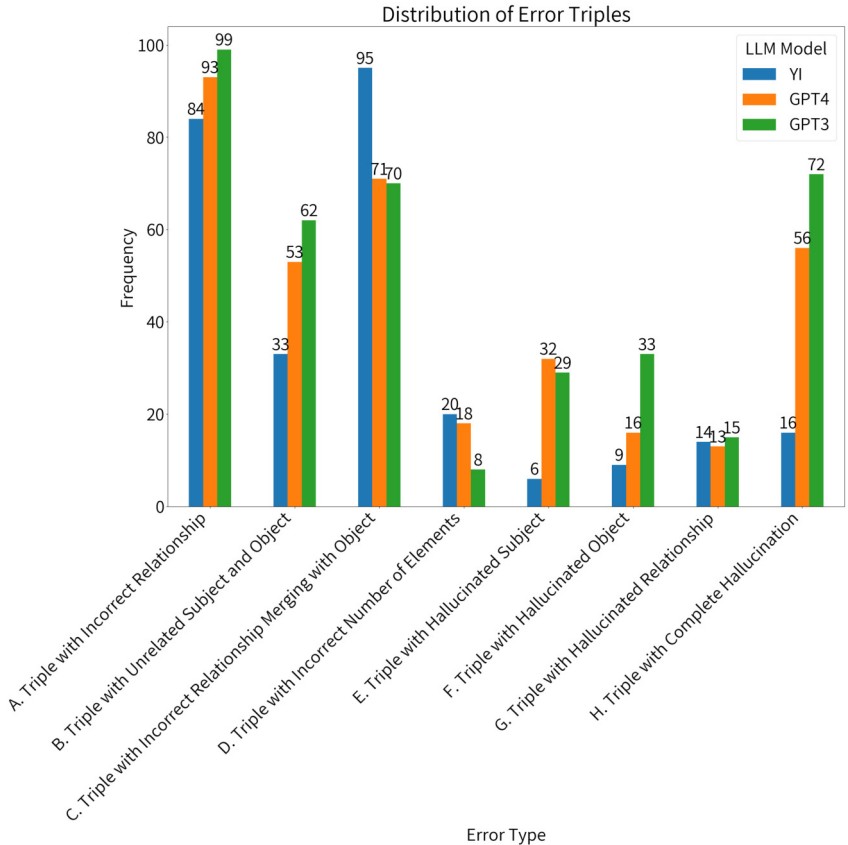

Figure 8: Distribution of Error Triples

## G Statistics of RQ3 Effectiveness and Case Study

### G.1 Characteristics of Knowledge Graph

Figure 9 and Figure 10 show the accuracy of the models for different text complexity. In the graph, there are 6301 nodes related to cyber security topics, categorized under "Attack Vector And Techniques", "Malware Name" and "CVE". Additionally, there are 4361 nodes related to other computer science topics, which include "Software/Tools", "File Type", "Domain Names", "Hashes", "IP Addresses" and "Email Addresses". The remaining 16365 nodes are classified under "Other", "Organization Name" and "Geographic Region". Overall, 39.45% of the nodes are related to computer science topics.. As for the edges, the graph includes 562 edges related to cyber security topics, 1355 edges related to other computer science topics, and a total of 1774 general edges. Overall, 51.94% of the edges pertain to computer science topics.

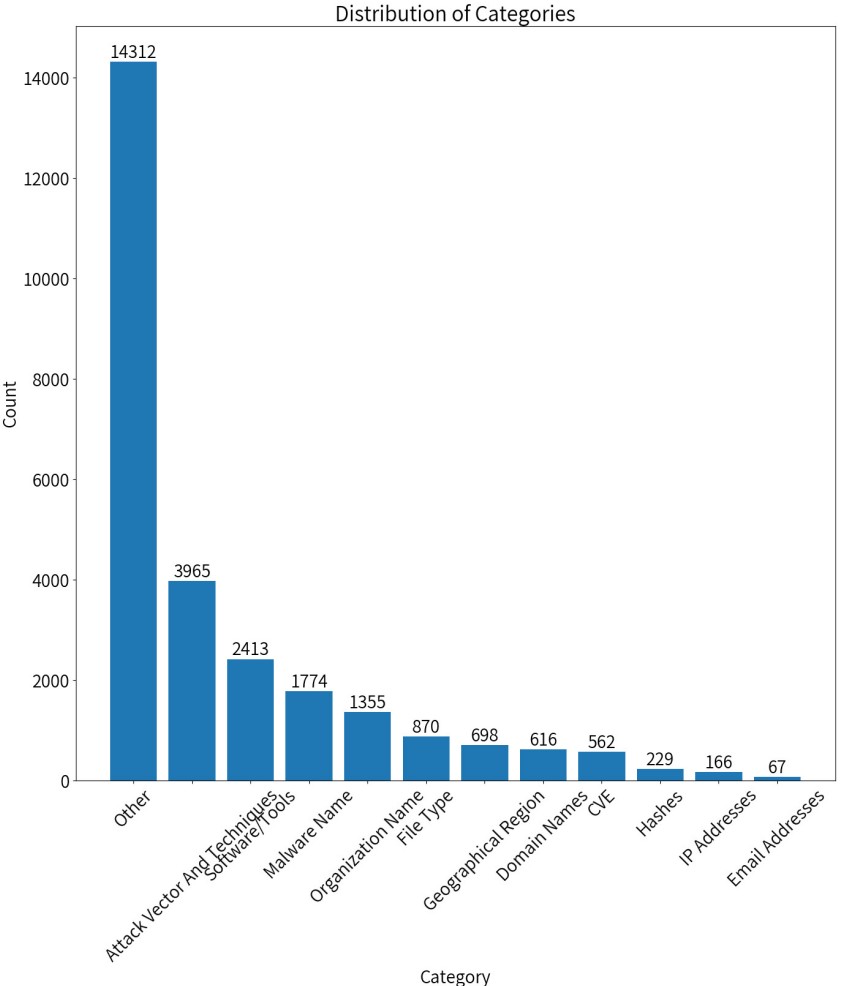

Figure 9: Distribution of Nodes Categories

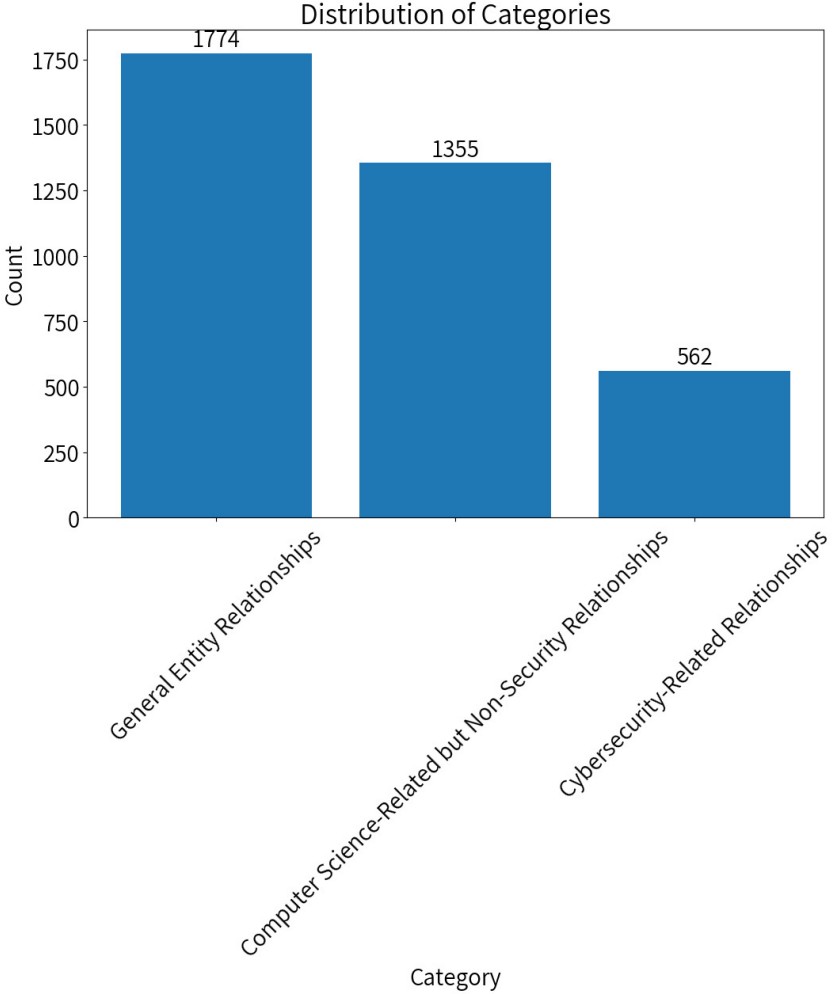

Figure 10: Distribution of Edges Categories

## G.2 Comparison of Knowledge Graph Effectiveness

Table 3 shows the effectiveness of cyber threat revelation specifically for each CVE.

Table 3: Effectiveness of CTIKG

| Name | # of Behaviors | Max # of Behaviors per Article | Added Behaviors | # of Articles | Avg Added Behaviors per Article |
|---|---|---|---|---|---|
| CVE-2021-44228 | 67 | 11 | 56 | 24 | 2.43 |
| CVE-2018-8174 | 67 | 9 | 58 | 21 | 2.9 |
| CVE-2012-0158 | 65 | 6 | 59 | 29 | 2.11 |
| CVE-2019-19781 | 60 | 9 | 51 | 18 | 3 |
| CVE-2017-11882 | 47 | 12 | 35 | 22 | 1.67 |
| CVE-2017-0199 | 45 | 10 | 35 | 15 | 2.5 |
| CVE-2010-1885 | 41 | 16 | 25 | 15 | 1.79 |
| CVE-2012-0507 | 33 | 5 | 28 | 14 | 2.15 |
| CVE-2021-34527 | 30 | 7 | 23 | 11 | 2.3 |
| CVE-2021-26855 | 29 | 11 | 18 | 11 | 1.8 |
| CVE-2010-2568 | 25 | 9 | 16 | 8 | 2.29 |
| CVE-2010-0188 | 24 | 2 | 22 | 21 | 1.1 |
| CVE-2021-26858 | 19 | 12 | 7 | 6 | 1.4 |
| CVE-2016-0189 | 19 | 6 | 13 | 11 | 1.3 |
| CVE-2018-4878 | 17 | 4 | 13 | 8 | 1.86 |

Column "# of Behaviors" shows the total number of behaviors of with a specific CVE, as identified from multiple articles. Column "Max # of Behaviors per Article" shows the highest number of behaviors identified in a single article. Column "Added Behaviors" shows the number of additional behaviors identified from other articles. Column "# of

Articles" shows the total number of articles that describe the behaviors of the specific CVE. Column "Avg Added Behaviors per Article" shows the average number of additional behaviors contributed by each article.

### G.3 Case Study

We identified two cases that illustrate how the knowledge graph reveals cyber threat intelligence. Figure 11 and Figure 12 show the correlations between the original article text and the corresponding knowledge graphs.

- *Reveal Hidden Information From Complex Text*: CTIKG can reveal information that does not exist in the NVD database by analyzing the CTI article. For example, the knowledge graph shows that CVE-2021-27065 is similar to CVE-2021-26858, and can overwrite any system file that isn't listed in NVD or other database.

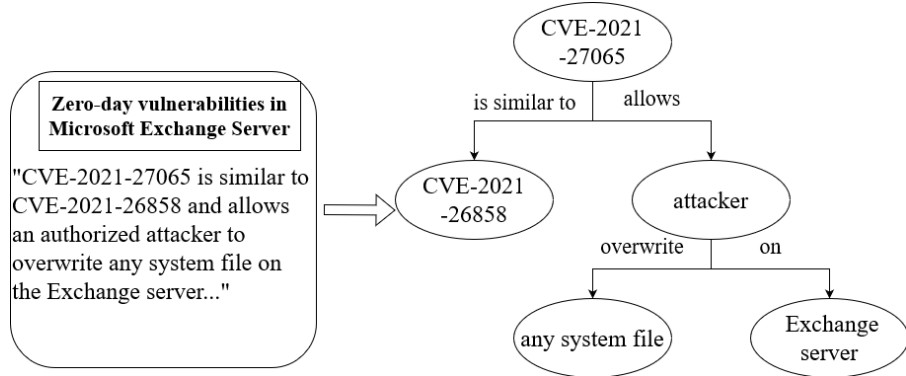

Figure 11: Cyber threat intelligence of CVE-2021-27065

- *Reveal Hidden Information From Multiple Articles*: CTIKG can reveal the path from one entity to another entity based on different articles. For example, the graph shows that both CVE-2012-0158 and CVE-2017-11882 are related to Microsoft Office, which informs the researcher who encounters the CVE-2017-11882 to check if the system is infected by CVE-2012-0158, since they both utilize Office.

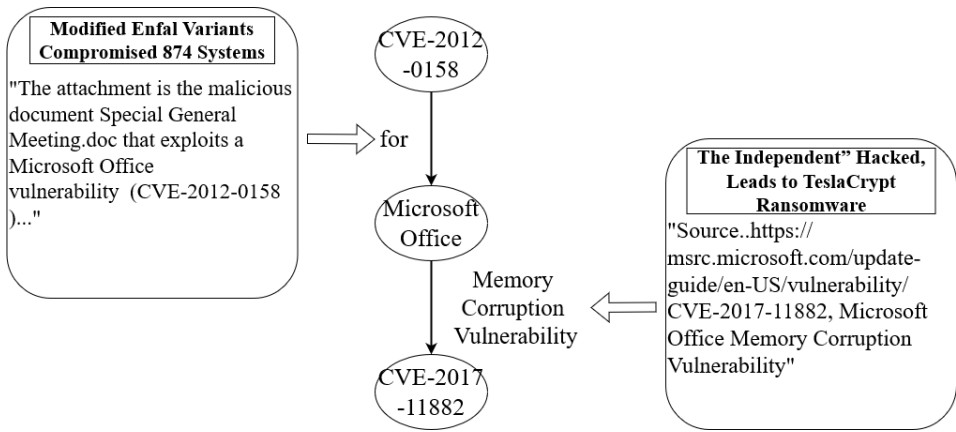

Figure 12: Cyber threat intelligence of CVE-2012-0158 and CVE-2017-11882

