# OpenReview forum: "CTIKG: LLM-Powered Knowledge Graph Construction from Cyber Threat Intelligence"
_colmweb.org/COLM/2024/Conference — COLM_

### Official Review · Reviewer_yYHM · 2024-04-29

**Rating:** 7
**Confidence:** 4
**Ethics Flag:** 1

**Summary:**

This paper presents an LLM-based system to extract triples and resolve them into a knowledge graph for a specific domain, namely cyber threat intelligence. The system involves a short-term memory constructor with multiple steps to create initial triples, a long-term memory constructor to merge and modify those triples, and finally the KG constructor.   In addition to the system itself, the paper presents a gold standard set of extracted triples in order to measure performance.  Several variants of the proposed system as well as several existing systems (both general KG construction and CTI specific) are compared against existing benchmarks, with the proposed system performing much better than any of the other solutions.

**Questions To Authors:**

In order of the paper.

Please have the paper proofread by a native speaker of English. There are issues with determiner placement, verb agreement and word usage.

Will the datasets and code (e.g. for evaluation) be released?  If so, state that and include under what license.

Sec 2: Add a sentence about how the system targets which entities and which relations. This becomes clearer later, but at this point it is confusing whether all subj-verb-obj triples are extracted and then later filtered out if not relevant.
Also add roughly how many unique nodes there are (e.g. 100, 1000, 100000) and how many unique relations (e.g. 10, 100, 1000).

Sec 3.2: What is IOC?
Checker Agent: Can you provide some rough indicator of how often the LLMs output incorrect results?  Is this 1%, 10% or 50% of the time?  Later in the paper with the error analysis (which is very helpful), if you have a rough estimate of how often each error type occurs, that would be interesting to include.

Sec 3.4: How does the system decide whether the node is a new one or should be merged with an existing one?  Often thresholding on similarity is hard (although if the node names are relatively the same length, this becomes easier) with embeddings and so some additional factor is needed.

sec 4.1: "sta" looks odd in the citation; use the name of the lab (e.g. Stanford NLP Lab or whatever the official name is)

Table 1: Include statistical significance of best result against next best.
Perhaps bold face the best results.
It would be great to know the number of triples of each type (row); could these be provided in parentheses after the name of the tactic?
Why does the GPT4 variant do better than CtiKG for Command & Control and Impact? Is there something different about these two compared to all the others?

Sec 4.3: In the long run, systems will have to overcome the coreference resolution issue if they are to work over textual documents. Any suggestions for future work for how to deal with this?

Table 2: Add statistical significance of best system over next best

Sec 6: Just merge this into the intro.  It will be more helpful there and this will save space for other analysis.

**Reasons To Accept:**

- Highlights the complexity of using LLMs for specific tasks, especially in domain-specific applications. This approach is likely to extend to many related domains.
- Provides a practical combination of techniques and is able to leverage LLMs in different ways. Includes use of standard NLP modules such as NER and text breaking.
- Interesting use of short and long term memory constructors with KG.
- Contains an error analysis for key components.

**Reasons To Reject:**

- The English is off, making the paper difficult to read due to distracting ungrammaticalities and odd phrasing.  This can be fixed by having the final paper proof-read by a native speaker of English.
- Needs to add statistical significance of results (given the major improvements, it is highly likely that these are statistically significant)

---

> ### Author Rebuttal · Authors · 2024-05-31
>
> Thank you for recognizing the novelty and the effectiveness of CTIKG.
>
> ## **Dataset and Code**
> As stated in the introduction, they have already been published on Github, the anonymous link is https://github.com/ctikgresearch/GTIKGResearch.
>
> ## **Writing Issues**
> Thanks for the suggestion. We will ask native English speakers to refine it, and solve other problems you mentioned in the review.
>
> ## **IOC**
> As stated in the introduction, IOCs are forensic artifacts of an intrusion, and we will add more details in the camera-ready version.
>
> ## **Error Analysis**
> As stated in Section 4.2, for the error rate, we found that even using GPT4 as LLM backend, the CTIKG still has 6.95% of error rate that can be detected and fixed by the checker. For the GPT3.5 model, the error will increase to 13.42%. More error analysis will be provided with case study in the camera-ready version.
>
> ## **Table 1 and 2**
> We will update the number of strategy types in Table 1 and provide the significance analysis in Tables 1 and 2 in the camera-ready version.
>
> ## **Node Merging**
> As stated in Section 3.4, CTIKG lists the five nodes that are most similar to the entities based on text embedding, and uses the merger agent to determine if these nodes refer to the same entities and merge them.
>
> ## **Command & Control and Impact**
> As stated in Section 4.2, CTIKG with GPT-4 demonstrates a superior text comprehension capability among all CTIKG variants. The texts in the Command & Control and Impact tactics exhibit greater diversities in writing, and our goal is to extract all entity relationships mentioned in the text, rather than based on the limited relationships of pre-defined ontology. In this case, CTIKG with GPT-4 can better utilize its comprehension capability to extract potential entity relationships, resulting in better accuracy.
>
> ## **Coreference Resolution**
> As stated in Section 3, CTIKG’s worker, refiner and merger agents are instructed to resolve coreference (prompts provided in Appendix.B). In the long run, we can add a new agent specifically for resolving coreference by consuming the whole article and collaborating with other agents.

---

> > ### Comment · Reviewer_yYHM · 2024-06-03
> >
> > Thank you for the clarifications. As mentioned in the review, this is very interesting research and it is great to know that the authors will be updating to fix some minor issues to make it easier to read.

---

> > > ### Author Response · Authors · 2024-06-03
> > >
> > > Thank you for taking time to evaluate the paper and your constructive feedback! Your feedback is greatly appreciated.

---

### Official Review · Reviewer_dKSc · 2024-05-10

**Rating:** 7
**Confidence:** 4
**Ethics Flag:** 1

**Summary:**

In this paper, the authors proposed a system that utilizes multiple LLM agents and a dual memory design to efficiently construct a knowledge graph from CTI articles. This advanced system breaks down lengthy CTI articles into smaller text segments, which are then processed individually by various LLM agents operating at different temperature settings. This approach serves to minimize the inherent randomness of LLMs. Following this segmented processing, the CTIKG summarizes the outcomes to produce more precise results. Evaluations conducted on three benchmarks, derived from actual CTI articles, have shown that CTIKG outperforms other cutting-edge methods.

Strengths:

1.The proposed multi-agent collaborative method for constructing knowledge graphs, as outlined in the paper, possesses practical value, particularly in the realm of knowledge representation and reasoning;

2.The paper is very well written with clarity, easy-to-follow logic, and adequate discussions;

3.After conducting extensive computational experiments, the results clearly demonstrated the effectiveness of the method.

Weaknesses:

1.The depth and breadth of the research conducted on knowledge graph construction(section 6) is disappointingly insufficient, lacking substantial content and comprehensive analysis;
2.In the process of collaboratively constructing the knowledge graph, for intelligent agents playing different roles, what are their contribution rates? Analysis results of this part of the content also need to be given.

**Questions To Authors:**

In the process of collaboratively constructing the knowledge graph, for intelligent agents playing different roles, what are their contribution rates?

**Reasons To Accept:**

1.The proposed multi-agent collaborative method for constructing knowledge graphs, as outlined in the paper, possesses practical value, particularly in the realm of knowledge representation and reasoning;

2.The paper is very well written with clarity, easy-to-follow logic, and adequate discussions;

3.After conducting extensive computational experiments, the results clearly demonstrated the effectiveness of the method.

**Reasons To Reject:**

1.The depth and breadth of the research conducted on knowledge graph construction (section 6) is disappointingly insufficient, lacking substantial content and comprehensive analysis;
2.In the process of collaboratively constructing the knowledge graph, for intelligent agents playing different roles, what are their contribution rates? Analysis results of this part of the content also need to be given.

---

> ### Author Rebuttal · Authors · 2024-05-31
>
> Thank you for recognizing the novelty and the effectiveness of CTIKG.
>
> ## **More Comprehensive Analysis**
> Due to page limit, we could not put in all the detailed analysis of knowledge graph construction in the submitted version, and will provide that in the Appendix of the camera-ready version. We have further analyzed CodeKGC, which is also an LLM-based approach for knowledge graphs. It can only extract pre-defined entity relationships and process a single sentence without considering the context of the whole article, while CTIKG can automatically uncover different types of security-oriented entity relationships and process the whole article with the help of dual memory design.
>
> ## **Contribution of Agents**
> As stated in the Approach section, each type of agents have their unique and indispensable task:
> * Multiple worker agents are used to extract diverse entity relationships with different temperature settings, mitigating the effect of randomness.Without them, CTIKG extracts fewer correct entity relationships due to randomness, resulting in a lower recall.
> * The integrator agent's role is to summarize the valid triples with identical meanings and output a representative triple. CTIKG without the integrator would produce redundant triples.
> * The refiner agent performs the normalization, resolution, and splitting of the composite triple with multiple entity relationships. The refiner agent assists the merger agent in unifying identical entities across multiple segments or articles. If the merger agent performs these tasks of the refiner during long-term memory construction, it would be constrained by token limitations and unable to produce complete results.
> * The checker agent detects common error patterns due to hallucination and activates the retry mechanism. As stated in Section 4.2,  even using the GPT4 as the backend, 6.95% of the results were affected by hallucination. For the GPT3.5 model, the rate of hallucination increases to 13.42%.
> *  The merger agent is responsible for merging the triples from the short-term memory to form the long-term memory and performs coreference resolution to merge the triples with identical entities, which is an indispensable function for short-term memory construction and long-term memory construction.

---

> > ### Comment · Reviewer_dKSc · 2024-06-05
> > **Rebuttal**
> >
> > Thank you for your answers that have solved my problem. I have no other questions.

---

> > > ### Author Response · Authors · 2024-06-06
> > >
> > > We are glad that our answers addressed your concerns and appreciate your feedback. Thank you again for your review and comments.

---

### Official Review · Reviewer_APiJ · 2024-05-11

**Rating:** 6
**Confidence:** 2
**Ethics Flag:** 1

**Summary:**

The paper introduces a novel approach, CTIKG, for constructing a security-oriented knowledge graph from Cyber Threat Intelligence (CTI) articles. The authors address the limitations of existing work and propose the use of LLMs to overcome these challenges. CTIKG employs prompt engineering and multiple LLM agents with a dual memory design to efficiently extract triples from the articles and generate results. The effectiveness of CTIKG is evaluated on three benchmarks constructed from real-world CTI articles, demonstrating significant improvements over state-of-the-art techniques.

**Questions To Authors:**

How scalable is your approach when working with larger-scale datasets or real-time applications? Did you encounter any performance issues during evaluation?

**Reasons To Accept:**

1. Novel approach. The paper addresses an important problem in cybersecurity by proposing a novel approach that leverages LLMs for knowledge graph construction from CTI articles.
2. Clear problem statement. The authors clearly identify the limitations of existing work and motivate the need for a specialized knowledge graph that captures relationships among security-related entities.
3. Robust evaluation. The paper provides comprehensive evaluations on multiple benchmarks, showcasing the effectiveness of CTIKG in terms of precision and recall compared to other approaches.

**Reasons To Reject:**

Limited discussion on generalizability. The authors focus solely on cybersecurity-related text analysis using LLMs, and it would be beneficial to discuss potential applications or extensions of their approach beyond this specific domain.

---

> ### Author Rebuttal · Authors · 2024-05-31
>
> Thank you for recognizing the novelty and effectiveness of CTIKG.
>
> ## **Scalability**
> In our evaluations, we demonstrate the effectiveness on the benchmark with an average length of 5200 characters. In particular, due to the dual-memory design, CTIKG can handle large documents by splitting them into segments as described in the Approach section and will not be limited by the document length.
>
> ## **Performance**
> We used YI-32B, the SOTA open source model before the submission, to process an article in 5 minutes on a server with dual RTX A6000 (equivalent to dual RTX 3090). Now the newly released model, LLaMA3-8B, has shown a better performance with a smaller model size according to LMSYS Chatbot Arena Leaderboard. Since CTIKG can seamlessly switch to any LLM model as the backend, in our new experiments, the new CTIKG with LLaMA3-8B only needs __50 seconds__ to process __a full article__ on a server with dual RTX 6000 ada (equivalent to dual RTX 4090). In the long run, the CTIKG with newer LLMs will achieve better processing performance with more advanced GPUs.
>
> ## **Other Domains**
> While it is not within our focused scope in the submitted version, in future work, we plan to verify the effectiveness of CTIKG with adaptation to other domain knowledge in other domains like SciERC.

---

### Official Review · Reviewer_hhCn · 2024-05-12

**Rating:** 6
**Confidence:** 3
**Ethics Flag:** 1

**Summary:**

The paper presents an approach using LLMs to automate the process of constructing a knowledge graph of cyber-security-related entities from text documents related to cyber threat intelligence. The approach is evaluated on a new, manually curated dataset of articles extracted from security-related websites and knowledge bases.

**Questions To Authors:**

I'm not sure I understood correctly the reason for not testing parameter-efficient fine-tuning methods such as LoRA. On page 9, the authors mention the inability of loading multiple LoRA in VLLM. But why would you need to load multiple LoRA simultaneously? Couldn't you (at least for the purpose of your experiments) perform each task one at a time, each time storing intermediate results ?

**Reasons To Accept:**

- the paper is clear and well-written
- the proposed approach seems sound and is evaluated on three distinct benchmarks (including an error analysis)
- the dataset employed for the evaluation is a nice contribution to the field (although it's unclear whether it will be released publicly)

**Reasons To Reject:**

- the scientific novelty of the approach is limited to the use of existing techniques for the use of LLMs in information extraction.
- as the paper focuses on the construction of knowledge graphs for the field of cyber-threat intelligence, it is unfortunate that the authors never actually describe in detail the types of entities and relations that they seek to extract.This should, in my view, feature much more prominently in the paper.
- the paper should include more information about the dataset of CTI articles employed for the evaluation. How was the annotation performed? If several annotators were involved, did you compute their inter-annotator agreement?

---

> ### Author Rebuttal · Authors · 2024-05-31
>
> ## **Novelty**
> As described in the introduction, directly using LLM in knowledge graph construction faces four challenges due to randomness and hallucination: _token limitation_, _incorrect triple format_, _incorrect output_, and _misunderstood task_. To address these challenges, we propose a novel design equipped with dual-memory design, multiple LLM agents, and retry mechanism. Dual-memory design overcomes the challenge of _token limitation_, multiple agents mitigate the challenges of _incorrect format_ and _incorrect output_ due to randomness, and retry mechanism improves the robustness by addressing the challenge of _misunderstood tasks_ caused by hallucination. CodeKGC, a knowledge graph construction approach also based on LLM, can only extract pre-defined entity relationships and handle each sentence independently without considering the context of other sentences in the article, while CTIKG can automatically uncover different types of security-oriented entity relationships and process the whole article with the help of dual memory design.
>
> ## **Entity Types and Relationships**
> As described in the introduction, unlike the otology-type knowledge graph that can only extract pre-defined relationships and entity types, our goal is to dynamically capture all types of relationships related to any cyber threats in CTI articles. We will add the statistics of unique entity types and relationships in the Appendix of the camera-ready version.
>
> ## **Dataset**
> The data was labeled by at least two authors. As stated in the introduction, we anonymously published the ground truth and the results of triple extraction and knowledge graph construction at https://github.com/ctikgresearch/GTIKGResearch.
>
> ## **LoRA Support**
> Regarding the switching of lora during processing, Vllm needs 1-2 minutes to load the model, which makes switching LoRA very time consuming. In addition, to maximize GPU utilization, we used multiple threads to process different articles at the same time, so they call different agents at the same time. In our recent development, we updated the model to the SOTA model LLaMA3, and we have successfully enabled CTIKG to support LoRA training, LoRA switching, and merging the trained LoRA back into the base model as a new model. The new code, LoRA file, and the new model will be updated in the Appendix of the camera-ready version.

---

> > ### Comment · Reviewer_hhCn · 2024-06-05
> >
> > Thank you for your answer. I agree that the article presents some useful technical solutions to extract triples from those CTI articles, notably to overcome token length limitations (by storing intermediate results) and ensure the formatting is correct. But those solutions are variants of techniques that are already commonly used by LLM practictioners.
> > I still believe that the article does not pay enough attention to the cyber-security aspects of their work. As stated in the abstract, the goal is to build a *security-oriented knowledge graph". There already exists dozens of such security-oriented KGs, many extrracted automatically through IE techniques (although not all LLM-based). Their purpose is often to allow for some automated reasoning and pattern discovery. To this end, it is often useful to define a priori the types of entities and relations one seeks to extract. In other words, for many applications of such KGs, restricting oneself to an ontology is often an advantage, not a limitation. I believe the article would benefit from a more thorough analysis of whether the extracted KG actually contain entities and relations that are useful and relevant from a cyber-security perspective.

---

> > > ### Author Response · Authors · 2024-06-06
> > >
> > > We truly appreciate your feedback and comments.
> > >
> > > ## **Security-oriented Knowledge Graph**
> > > Regarding the security-oriented knowledge graph, we have implemented the following designs to ensure the security orientation. We specified "Only extract triples that are related to cyber attacks. If a sentence does not contain any triple about cyber attacks, skip the sentence and do not include it in your output" and "Focus on malware, Trojan horses, CVEs, or hacking organizations as the subjects of the triples'' in the prompts of LLM agents, and provided multiple security-related examples, such as "[Leafminer, attempts to infiltrate, target networks]" in the few-shot learning section of LLM agents. According to the evaluation results, CTIKG with these designs can be guided to focus on security-related entities and their relationships.
> > >
> > > ## **Related IE Approach**
> > > Regarding related IE approaches, we compared CTIKG with Extractor, the SOTA non-LLM based IE approach specifically designed for cyber threat report text. In the evaluation, CTIKG outperforms it in both sentence triple extraction and article knowledge graph construction. For sentence triple extraction, CTIKG outperforms Extractor by about 9% in both precision and recall. For article knowledge graph construction, CTIKG outperforms Extractor by about 30% in both precision and recall.
> > >
> > > ## **Type of  Entity Relationship**
> > > Regarding the ontology of knowledge graph, first, the article sources are from different websites and authors. These authors have very different writing styles, and pose challenges for a fixed ontology to represent the entities and their relationships. To address this challenge, CTIKG adopts multiple agents to unify different writing styles and achieve better coreference resolution (such as the refiner on the triple level and the merger agent on the segment and article levels). Extractor, on the other hand, suffers from low precision due to this challenge. Second, non-security-related relationships between security-related entities also play an indispensable role for connecting correlated entities, such as two security-related entities like CVEs described in different sentences but connected via a non-security-related entity. In our Appendix case study, the similarity between CVE-2012-0158 and CVE-2017-1188 was discovered based on the entity relationship of “for” and “memory corruption vulnerability”. Our further investigation revealed that Magniber and Cerber have a "mutual ransomware payload" relationship, and Emotet and Trickbot have a "switch to" relationship.
> > > These entity relationships are determined by the author's writing style and cannot be anticipated in advance and defined in the ontology. We will include the effectiveness analysis of non-security-related relationships in the Appendix of the camera-ready version.

---

### Decision · Program_Chairs · 2024-07-10

**Decision:**

Accept

**Comment:**

The paper is a solid investigation of the application of LLMs to a realistic information extraction setup, dealing with issues such as hallucinations and context limitations. The evaluation is thorough. The reviewers agreed on these points and the authors addressed their questions in their rebuttals adequately.